# Platinum Plus Tegafur–Uracil versus Platinum Alone during Concurrent Chemoradiotherapy in Patients with Nonmetastatic Nasopharyngeal Carcinoma: A Propensity-Score-Matching Analysis

**DOI:** 10.3390/cancers14184511

**Published:** 2022-09-17

**Authors:** Ching-Feng Lien, Chien-Chung Wang, Chuan-Chien Yang, Chih-Chun Wang, Tzer-Zen Hwang, Yu-Chen Shih, Shyh-An Yeh, Meng-Che Hsieh

**Affiliations:** 1Department of Otolaryngology, E-Da Hospital, Kaohsiung 824005, Taiwan; 2College of Medicine, I-Shou University, Kaohsiung 84001, Taiwan; 3Department of Otolaryngology, E-Da Cancer Hospital, Kaohsiung 82445, Taiwan; 4Department of Radiation Oncology, E-Da Hospital, Kaohsiung 82445, Taiwan; 5Department of Hematology-Oncology, E-Da Cancer Hospital, Kaohsiung 82445, Taiwan

**Keywords:** tegafur–uracil, chemoradiotherapy, nasopharyngeal carcinoma, prognosis, survival

## Abstract

**Simple Summary:**

The current guidelines suggest that concurrent chemoradiotherapy (CCRT) with a cisplatin-based regimen is the standard treatment for patients with nonmetastatic NPC. However, little is known regarding the addition of tegafur–uracil (UFUR) to CCRT in patients with NPC. Our study showed that the median RFS was not reached (NR) in the CCRT-UP group, and it was 12.5 months in the CCRT-P group (*p* < 0.001). The median OS was NR in the CCRT-UP group, and it was 15.9 months in the CCRT-P group (*p* < 0.001). Grade 3–4 AEs were insignificant between the CCRT-UP and CCRT-P arms. In summary, CCRT-UP had better RFS and OS in nonmetastatic NPC patients with similar toxic profiles.

**Abstract:**

Concurrent chemoradiotherapy (CCRT) with a cisplatin-based regimen is the standard treatment for patients with nasopharyngeal carcinoma (NPC). Our study was a propensity-score-matching analysis and it aimed to investigate the oncologic outcomes of platinum plus tegafur–uracil versus platinum alone during CCRT in patient with nonmetastatic NPC. Patients with pathologic confirmed NPC in 2018–2022 were reviewed. Patients treated with platinum plus tegafur–uracil (CCRT-UP) or platinum alone (CCRT-P) during CCRT were recruited into this study. A propensity-score-matching analysis was conducted to diminish the selection bias. The recurrence-free survival (RFS) and overall survival (OS) were presented with Kaplan–Meier curves. The treatment-related adverse effects (AEs) were recorded according to the National Cancer Institute’s Common Terminology Criteria V3.0. A total of 44 patients with CCRT-UP and 44 patients with CCRT-P were identified after propensity score matching. The median RFS was not reached (NR) in the CCRT-UP group, and it was 12.5 months in the CCRT-P group (*p* < 0.001). The median OS was NR in the CCRT-UP group, and it was 15.9 months in the CCRT-P group (*p* < 0.001). The overall response rate and disease-control rate were insignificant between the CCRT-UP and CCRT-P groups. A subgroup analysis showed that the median OS was significantly longer in the CCRT-UP group than in the CCRT-P group, regardless of the clinical stage. A multivariate analysis exhibited that CCRT-UP was independently correlated with survival. The grade 3–4 AEs were insignificant between the CCRT-UP and CCRT-P arms. CCRT-UP had better RFS and OS in nonmetastatic NPC patients with similar toxic profiles. Further larger-scaled prospective randomized control trials are warranted to validate our conclusions.

## 1. Introduction

Nasopharyngeal carcinoma (NPC) is the most common malignant neoplasm arising from lymphocyte-rich nasopharyngeal mucosa [1]. NPC has a unique pathogenesis and is distinct from other head and neck cancers. The main etiologies of NPC includes Epstein–Barr virus (EBV) infection, as well as genetic alterations and diet habits [2,3].

The distribution of NPC demonstrates a clear regional, racial and gender prevalence [4]. Although NPC is relatively uncommon in Western countries, it remains a common malignancy in Asian countries. There were approximately 129,000 cases with new diagnoses of NPC worldwide in 2018, and more than 70% of these patients were diagnosed in Asia [5]. The age-standardized incidence rates of NPC were different between Asia and other countries, accounting for 5 per 100,000 persons in Southeast Asia, and 1.6 per 100,000 persons in the world [6]. Previous literature reviews have all demonstrated that the incidences of NPC were higher in men than in women, with a ratio of 2–3:1 [7]. Moreover, age, gender and histological subtypes were correlated with survival. The impact of age played an important role in survival. For patients aged 15–45 years, the 5-year survival rate was 72%, while for patients aged 65–74 years, the 5-year survival rate was 36% [8]. As for gender differences, female patients usually have better prognoses than male patients [9,10]. The NPC-related mortality rates of the different histological subtypes were also significantly different [11].

The early diagnosis of NPC is difficult because of its anatomical location and nonspecific symptoms. [12]. According to the 7th Edition of the American Joint Commission on Cancer staging system, two-third of NPC patients were diagnosed as having locally advanced disease, and they were at high risk for local recurrence as well as distant metastases [13]. The current guidelines suggest that concurrent chemoradiotherapy (CCRT) with a cisplatin-based regimen is the standard treatment for patients with nonmetastatic NPC [14]. The cumulative cisplatin dose (CCD) during CCRT was demonstrated to be prognostic for patients with NPC [15]. Peng et al. evaluated the threshold of CCD and showed that 240 mg/m^2^ is optimal for NPC patients treated with CCRT. Chan et al. conducted a phase III randomized study to compare the oncologic outcomes between CCRT and radiotherapy (RT). The results revealed that the 5-year survival rates were apparently 70% and 58% in the CCRT and RT groups, respectively [16]. However, many NPC patients have residual tumors after chemoradiotherapy and easily develop distant metastases in the near future. Thus, the treatment intensity of CCRT with cisplatin monotherapy may be insufficient for patients with high-risk NPC. Oral tegafur–uracil (UFUR) (TTY Biopharm, Taiwan) consists of a 1:4 molar ratio of tegafur and uracil. Tegafur is an oral prodrug of fluoropyrimidine, and it can be converted into 5-fluorouracil (5-FU) in vivo. Uracil is an oral inhibitor of dihydropyrimidine dehydrogenase, which can enhance the efficacy of 5-FU without increasing its toxicity. Previous studies have confirmed its role in the maintenance therapy of NPC [17,18]. However, little is known regarding the addition of UFUR to CCRT in patients with NPC. Our study was a propensity-score-matching analysis, and it aimed to investigate the oncologic outcomes of platinum with or without UFUR during CCRT in patient with nonmetastatic NPC.

## 2. Materials and Methods 

### 2.1. Patients 

NPC patients treated with definitive CCRT as a curative intent from 2018 to 2022 at E-Da Hospital were retrospectively reviewed. Patients who were aged 18–75 years, had pathologically confirmed nonmetastatic NPC and had received platinum with or without UFUR during CCRT were recruited into our study. The exclusion criteria included metastatic disease, palliative CCRT, a surgical resection of primary tumor or neck lymph nodes before CCRT, an incomplete scheduled CCRT course, other chemotherapy regimens during CCRT and irregular follow-up intervals. Patients were classified into the “CCRT-UP” group if they received platinum plus UFUR during CCRT, followed by UFUR maintenance for 1 year, or until disease progression or unacceptable toxicity. Patients were classified into the “CCRT-P” group if they received platinum alone during CCRT, followed by the best supportive treatment until disease progression or unacceptable toxicity. Neoadjuvant and adjuvant chemotherapies were prohibited in this study to reduce the influences of other chemotherapies. The basic characteristics of our patients were retrospectively collected from a medical-chart review. In order to reduce the selection bias, a propensity-score-matching analysis was conducted for the oncologic comparison. This study was approved by the E-Da Hospital Institutional Review Board (EMPR70110N), and it was conducted in accordance with the Declaration of Helsinki. Our study was a retrospective study, which was exempt from requiring consent.

### 2.2. Treatment Methods

All patients underwent conventional radiotherapy with concurrent platinum chemotherapy according to our treatment guidelines. Patients received 2.00 Gy per fraction of radiotherapy, with a total of 70 Gy in 35 fractions. The planned dose of the gross tumor volume was 70 Gy, and for the lymph nodes it was 70 Gy. The clinical tumor volume was 60 Gy, and the secondary tumor volume was 54 Gy. The radiotherapy was administrated daily for 7 weeks. For the CCRT-UP, patients received weekly cisplatin at 35 mg/m^2^ or carboplatin AUC 2 on day 1, and oral UFUR 200 mg twice per day plus oral leucovorin 100 mg twice per day during CCRT, followed by UFUR plus leucovorin for 1 year, or until disease progression or intolerance. For CCRT-P, patients received weekly cisplatin at 35 mg/m^2^ or carboplatin AUC 2 on day 1 during CCRT, followed by the best supportive care until disease progression. Patients with fit renal function were treated with cisplatin, while patients with unfit renal function, peripheral neuropathy and hearing impairment were treated with carboplatin. Computed tomography or magnetic resonance imaging were scheduled to evaluate the treatment response 1 month after CCRT, and every 2–3 months in the following days. 

### 2.3. Statistical Analysis

All baseline data of our patients were retrieved by medical-chart review. Chi-square tests were used to compare both groups. Statistical analyses were performed with SPSS. In order to reduce the selection bias, patients with CCRT-UP were matched with those with CCRT-P by propensity score matching (PSM) according to age, gender, performance status, renal function and clinical stage. The oncologic outcomes were presented with recurrence-free survival (RFS), overall survival (OS), the overall response rate (ORR) and the disease-control rate (DCR). RFS refers to the interval from diagnosis to tumor recurrence or the final follow-up or death, while OS refers to the interval from diagnosis to death or the final follow-up. The evaluations of the objective responses were measured according to the RECIST 1.1 criteria, with complete response (CR), partial response (PR), stable disease (SD), and progressive disease (PD). The ORR was defined as the CR plus the PR, and the DCR was defined as the CR, the PR, plus the SD. Kaplan–Meier curves were estimated with the survival using a log-rank test. Cox regression analysis was also conducted to adjust the potential confounders with the “enter” selection. P values were considered to be significant if they were <0.05. Treatment-related adverse events (AEs) were documented according to the National Cancer Institute’s Common Terminology Criteria (NCICTC) V3.0.

## 3. Results

### 3.1. Patient Characteristics 

There were 118 patients recruited in this study to compare the oncologic outcomes. The median age of our patients was 53 years. Table 1 summarizes the baseline characteristics of our patients. Generally, 73% were male patients, and 63% were younger patients with ages less than 60 years. More than 90% of the patients had Eastern Cooperative Oncology Group performance status (ECOG PS) 1, and 74% of the patients had fit renal function. Most of the patients were diagnosed with an advanced stage: 63% of the patients had clinical stage IVA, and 37% patients had stage II–III disease. After the propensity score matching, a total of 44 patients in the CCRT-UP group, and 44 patients in the CCRT-P group, were recruited to compare the oncologic outcomes. All the basic characteristics, including gender, age, ECOG PS, renal function and initial clinical stage, were balanced between these two treatment arms. 

### 3.2. Survival Outcomes 

The median follow-up period was 13.1 months. On the cuff-off date of our study, 26 (30%) patients died, and cancers were the major causes of their deaths. Table 2 presents the oncologic outcomes between the CCRT-UP and CCRT-P groups. The median RFS was not reached (NR) in the CCRT-UP group, and it was 12.5 months in the CCRT-P group (*p* < 0.001). The median OS was NR in the CCRT-UP group, and it was 15.9 months in the CCRT-P group (*p* < 0.001). Figure 1 plots the Kaplan–Meier curves of the RFS and OS. The ORR and DCR were both insignificant, accounting for 91% versus 87% (*p* = 0.368), respectively, and 93% versus 92% (*p* = 0.773), respectively, in the CCRT-UP and CCRT-P groups, respectively. Then, we stratified our patients according to their clinical stages. The OS curves of stage II–III and stage IVA are plotted in Figure 2. The median OS was significantly longer in the CCRT-UP group than in the CCRT-P group, regardless of the clinical stage. For patients with clinical stage II–III disease, the median OS was NR in the CCRT-UP group, and it was 15.9 months in the CCRT-p group (*p* = 0.001). For patients with clinical stage IVA disease, the median OS was NR in the CCRT-UP group, and it was 14.5 months in the CCRT-P group (*p* = 0.005). Table 3 presents the results of the Cox regression analyses with the survival. A multivariate analysis demonstrated that CCRT-UP was an independently positive predictor for the RFS and OS. 

### 3.3. Safety Profiles

In general, the AEs were insignificantly different between the CCRT-UP and CCRT-P arms. Table 4 presents all the grade 3–4 AEs in our study. Overall, the grade 3–4 AEs in the CCRT-UP group were numerically higher than those in the CCRT-P group, accounting for 11 (25%) versus 9 (20%), respectively (*p* = 0.313). As for ≧grade 3, hematologic toxicity, neutropenia and anemia were the major AEs in our study. In summary, four (9%) patients in the CCRT-UP group and three (7%) patients in the CCRT-P group had ≧grade 3 neutropenia (*p* = 0.566), while three (7%) patients in the CCRT-UP group and two (5%) patients in the CCRT-P group had ≧grade 3 anemia (*p* = 0.514). As for ≧grade 3, nonhematologic toxicity, vomiting, anorexia and oral mucositis were the most common AEs in our study. In summary, six (14%) patients in the CCRT-UP group and five (11%) patients in the CCRT-P group had ≧grade 3 vomiting (*p* = 0.423), while four (9%) patients in the CCRT-UP group and five (11%) patients in the CCRT-P group had ≧grade 3 anorexia (*p* = 0.589), while seven (16%) patients in the CCRT-UP group and five (11%) patients in the CCRT-P group had ≧grade 3 oral mucositis (*p* = 0.257). Furthermore, three (7%) patients in the CCRT-UP group had fatigue, and three (7%) in the CCRT-P group developed peripheral neuropathy. 

## 4. Discussion

To the best of our knowledge, this is the first study that compares the oncologic outcomes between CCRT-UP and CCRT-P in nonmetastatic NPC. The current guidelines suggest that CCRT with a cisplatin-based regimen is the standard treatment for patients with NPC [14]. Our study showed that CCRT-UP was superior to CCRT-P in patients with nonmetastatic NPC in terms of the RFS and OS (*p* < 0.001). The ORR and DCR were both insignificant in the CCRT-UP and CCRT-P groups. The safety profiles were also similar between the CCRT-UP and CCRT-P groups. After stratifying our patients according to their clinical stages, CCRT-UP remained more effective, regardless of the clinical stage. Given that the patient number was small in this study, our conclusions have clinical implications for physicians who treat patients with NPC. Further larger-scaled prospective randomized control trials are warranted to validate our conclusions.

According to the National Comprehensive Cancer Network guidelines, the treatment of NPC is based on the clinical stage. [14]. CCRT is adopted to treat patients with nonmetastatic NPC [2]. Previous literature published by the Meta-Analysis of Chemotherapy in Nasopharynx Carcinoma (MAC-NPC) group disclosed that CCRT had an absolute 5-year survival benefit of 6.3% [19]. The most common regimen for NPC patients with CCRT is cisplatin-based treatment [20]. Zhu et al. investigated the efficacy and acute toxicities of CCRT from weekly low-dose cisplatin (40 mg/m^2^) to triweekly high-dose cisplatin (100 mg/m^2^) [21]. They concluded that the weekly low-dose cisplatin regimen during CCRT resulted in significantly better distant-metastases-free survival in patients with locally advanced NPC. In terms of the AEs, CCRT with a weekly low-dose cisplatin regimen exhibited significantly more grade 3–4 hematologic toxicity, but less grade 3–4 gastrointestinal toxicity, than the triweekly high-dose cisplatin group. A systematic review and meta-analysis also focused on the comparison between the triweekly high-dose cisplatin and weekly low-dose cisplatin [22]. The treatment efficacy in terms of the OS and ORR were similar between the weekly low-dose and triweekly high-dose cisplatin regimens during CCRT. The weekly regimen was more compliant and significantly less toxic with respect to severe AEs. Based on these results, we prefer the weekly cisplatin regimen during CCRT for patients with nonmetastatic NPC in clinical practice. Thus, the weekly cisplatin-based regimen was adopted in this study. 

Taiwan is a pandemic area of NPC and Epstein–Barr virus infection. The treatment outcomes of cisplatin alone during chemoradiotherapy in Taiwan are miserable, and most of our NPC patients had N3 or stage IVA when they were newly diagnosed. These patients were at great risks to develop distant metastases, which led to poor survival in our study. Thus, more intensified chemotherapy regimens during chemoradiotherapy are warranted to improve the treatment efficacy. Previous studies have investigated cisplatin-based doublet regimens during CCRT in patients with NPC. Lin et al. published a phase III study regarding CCRT with cisplatin and 5-FU (CF regimen) versus RT in patients with NPC. The 5-year survival rates were significantly different between the CCRT and RT arms, accounting for 72.3% and 54.2%, respectively (*p* = 0.0022) [23]. However, a higher incidence of stomatitis was found with the CF regimen, leading to more hospitalization and treatment interruption [24]. Yan et al. performed a multicenter open-label randomized study to investigate the treatment effectiveness of CCRT with raltitrexed (Saiweijian^®^) plus cisplatin (SP regimen) versus CCRT with 5-fluorouracil plus cisplatin (FP regimen) among locally advanced NPC patients. The 3-year PFS rates were 70.1% and 66.6% for SP and FP, respectively, and the 3-year OS rates were 84.0% and 73.7% for the SP and FP groups, respectively [25]. These cisplatin-based doublet regimens achieved better PFS and OS, but the toxicity raised more concerns during the CCRT. In this study, we used a novel combination regimen of platinum plus UFUR during the CCRT. Our results disclosed better RFS and OS toward CCRT-UP in comparison with CCRT-P among the nonmetastatic NPC patients. Moreover, it is worth noting that the AEs of CCRT-UP were similar to those of CCRT-P. 

Some regimens other than cisplatin were also proposed with CCRT in patients with NPC. Li et al. compared the oncologic outcomes of nimotuzumab versus cisplatin during CCRT of patients with locally advanced NPC, and they found that the 5-year OS and PFS rates were 63.9% versus 81.4% (*p* = 0.024) and 58.0% versus 80.6% (*p* = 0.028) for nimotuzumab and cisplatin, respectively [26]. However, less AEs were reported in the nimotuzumab arm, including leukopenia, nausea and vomiting. Another cost-effective analysis between nedaplatin-based CCRT and cisplatin-based CCRT showed that nedaplatin could be an alternative regimen during CCRT in NPC patients with stage II–IVB disease [27]. The outcomes of these were inferior to the standard cisplatin regimen in terms of the PFS and OS, which might limit the usage of these regimens in patients with NPC. However, for cisplatin-ineligible patients, these regimens could be one of the treatment choices.

Given the retrospective design, our study has several potential limitations. First, this was not a randomized control study, which might be the major bias. The chemotherapy regimen was determined at the physician’s discretion. Second, the single institutional experience with a small patient cohort and short follow-up periods may also restrict the power of our study. However, this study compared the efficacy and safety between CCRT-UP and CCRT-P in patients with nonmetastatic NPC. Our results disclosed a better survival of CCRT-UP in comparison with CCRT-P. Although this was a retrospective study with several inevitable selection biases, our conclusion could have clinical implications for physicians who treat patients with NPC.

## 5. Conclusions

In this propensity-score-matching analysis, we compared the oncologic outcomes of CCRT-UP with those of CCRT-P in patients with nonmetastatic NPC. Based on our results, we disclosed that CCRT-UP had better RFS and OS than CCRT-P. The ORR and DCR of CCRT-UP and CCRT-P were similarly insignificantly different. Regardless of the clinical stage, the median OS was consistently longer for CCRT-UP than CCRT-P. In our multivariate analysis, CCRT-UP was a strong prognostic factor related to the RFS and OS. Furthermore, the safety profiles were also similar between the CCRT-UP and CCRT-P arms. Our conclusion is real-world evidence and could have clinical implications for physicians who treat patients with NPC. Further larger-scaled prospective randomized control trials are warranted to validate our conclusions.

## Figures and Tables

**Figure 1 cancers-14-04511-f001:**
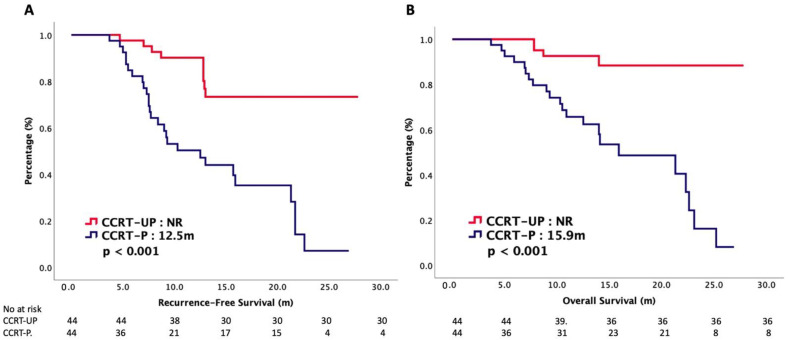
(**A**) Recurrence-free survival and (**B**) overall survival of patients with nonmetastatic nasopharyngeal carcinoma treated with concurrent chemoradiotherapy, stratified by chemotherapy regimen.

**Figure 2 cancers-14-04511-f002:**
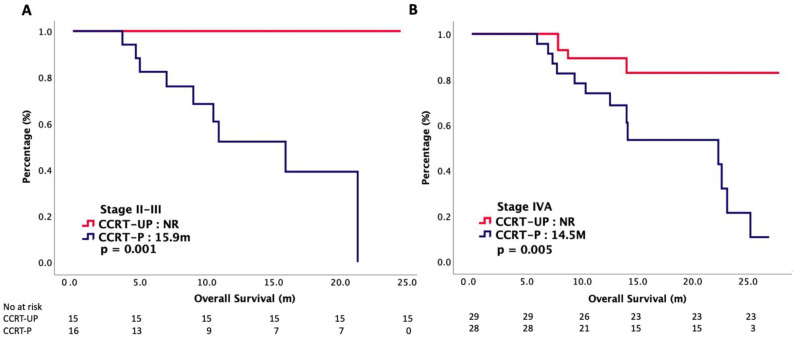
Overall survival of patients with nonmetastatic nasopharyngeal carcinoma treated with concurrent chemoradiotherapy, stratified by stage and chemotherapy regimen: (**A**) stage II–III; (**B**) stage IVA.

**Table 1 cancers-14-04511-t001:** Baseline characteristics of patients with nasopharyngeal carcinoma before and after PSM, stratified by chemotherapy regimens.

Variables	Before PSM			After PSM		
	CCRT-UP	CCRT-P	*p*	CCRT-UP	CCRT-P	*p*
	N = 44	N = 74		N = 44	N = 44	
Gender			0.376			0.999
Male	30 (68%)	56 (76%)		30 (68%)	30 (68%)	
Female	14 (32%)	18 (24%)		14 (32%)	14 (32%)	
Age			0.082			0.999
≦60 years	32 (73%)	42 (57%)		32 (73%)	32 (73%)	
>60 years	12 (27%)	32 (43%)		12 (27%)	12 (27%)	
ECOG PS			0.952			0.999
0–1	41 (93%)	67 (91%)		41 (93%)	41 (93%)	
2	3 (7%)	7 (9%)		3 (7%)	3 (7%)	
Renal function			0.211			0.697
CCr ≥ 60 mL/min	36 (82%)	51 (69%)		36 (82%)	34 (77%)	
CCr < 60 mL/min	8 (18%)	23 (31%)		8 (18%)	10 (23%)	
T stage			0.633			0.851
T1–T2	24 (55%)	37 (50%)		24 (55%)	23 (52%)	
T3–T4	20 (45%)	37 (50%)		20 (45%)	21 (48%)	
N stage			0.481			0.892
N0–N1	11 (25%)	23 (31%)		11 (25%)	10 (23%)	
N2–N3	33 (75%)	51 (69%)		33 (75%)	34 (77%)	
Stage			0.615			0.869
II–III	15 (34%)	29 (39%)		15 (34%)	16 (36%)	
IVA	29 (66%)	45 (61%)		29 (66%)	28 (64%)	

CCRT-UP: concurrent chemoradiotherapy with platinum plus tegafur–uracil; CCRT-P: concurrent chemoradiotherapy with platinum alone; PSM: propensity score matching; ECOG PS: Eastern Cooperative Oncology Group (ECOG) performance status; CCr: clearance of creatinine.

**Table 2 cancers-14-04511-t002:** Oncologic outcomes of patients with nasopharyngeal carcinoma, stratified by chemotherapy regimen.

Oncologic Outcomes	CCRT-UP N = 44	CCRT-P N = 44	*p*
mRFS (m)	NR	12.5	< 0.001
mOS (m)	NR	15.9	< 0.001
CR (%)	17 (39)	13 (30)	
PR (%)	23 (52)	25 (57)	
SD (%)	1(2)	2 (5)	
PD (%)	3 (7)	4 (8)	
ORR (%)	40 (91)	32 (87)	0.368
DCR (%)	41 (93)	38 (92)	0.773

CCRT-UP: concurrent chemoradiotherapy with platinum plus tegafur–uracil; CCRT-P: concurrent chemoradiotherapy with platinum alone; mRFS: median recurrence-free survival; mOS: median overall survival; NR; not reached; CR: complete response; PR: partial response; SD: stable disease; PD: progressive disease; ORR: objective response rate; DCR: disease-control rate.

**Table 3 cancers-14-04511-t003:** Multivariate Cox regression analysis of parameters associated with survival in patients with nasopharyngeal carcinoma.

Variables	RFS		OS	
	HR (95% CI)	*p* Value	HR (95% CI)	*p* Value
Gender (female vs. male)	0.81 (0.38–1.72)	0.574	0.64 (0.24–1.69)	0.362
Age (≦60 years vs. >60 years)	0.68 (0.33–1.40)	0.300	0.74 (0.31–1.79)	0.499
ECOG PS (0–1 vs. 2)	0.78 (0.24–1.85)	0.420	0.87 (0.21–1.84)	0.515
Renal function (CCr ≥ 60 vs. < 60)	0.49 (0.21–1.23)	0.211	0.45 (0.17–1.21)	0.114
T stage (T1–T2 vs. T3–T4)	0.82 (0.42–1.59)	0.561	0.79 (0.36–1.73)	0.549
N stage (N0–N1 vs. N2–N3)	0.75 (0.21–2.71)	0.666	0.71 (0.19–2.73)	0.619
Clinical stage (stage II–III vs. IVA)	0.57 (0.27–1.22)	0.150	0.86 (0.38–1.96)	0.721
CCRT (CCRT-UP vs. CCRT-P)	0.21 (0.09–0.48)	<0.001	0.17 (0.06–0.52)	0.002

RFS: recurrence-free survival; OS: overall survival; HR: hazard ratio; CI: confidence interval; ECOG PS: Eastern Cooperative Oncology Group performance status; CCr: clearance of creatinine; CCRT-UP: concurrent chemoradiotherapy with platinum plus tegafur–uracil; CCRT-P: concurrent chemoradiotherapy with platinum alone.

**Table 4 cancers-14-04511-t004:** Grade 3 to 4 treatment-related adverse events in patients with nasopharyngeal carcinoma, stratified by chemotherapy regimen.

Adverse Events	CCRT-UP N = 44	CCRT-P N = 44	*p*
Hematologic events (*n* (%))			
Neutropenia	4 (9)	3 (7)	0.566
Febrile neutropenia	1 (2)	1 (2)	0.999
Anemia	3 (7)	2 (5)	0.514
Nonhematologic events (*n* (%))			
Skin rash	2 (5)	2 (5)	0.999
Fatigue	3 (7)	2 (5)	0.523
Diarrhea	2 (5)	2 (5)	0.999
Vomiting	6 (14)	5 (11)	0.423
Anorexia	4 (9)	5 (11)	0.589
Oral mucositis	7 (16)	5 (11)	0.257
Hand–foot syndrome	1 (2)	1 (2)	0.999
Peripheral neuropathy	2 (5)	3 (7)	0.549

## Data Availability

The deidentified data supporting this manuscript are available upon reasonable request to the corresponding author.

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
