# Peer review of "Platinum Plus Tegafur–Uracil versus Platinum Alone during Concurrent Chemoradiotherapy in Patients with Nonmetastatic Nasopharyngeal Carcinoma: A Propensity-Score-Matching Analysis"

_cancers, 2022, doi:10.3390/cancers14184511_

Round 1

Reviewer 1 Report

Because of  some sections detect plagiarism,it would be better if the author can rewrite the method part.

Author Response

Thanks for your kind review. Here are our responses.

  1. Because of some sections detect plagiarism, it would be better if the author can rewrite the method part

Thanks for your review. We had revised the method part.

Reviewer 2 Report

The authors of the article present a retrospective analysis of non-metastatic NPC patients treated either with platinum alone or combined with UFUR. They found statistically significant better OS and PFS for the combination treatment. 

The study ist of clinical interest as NPC are primarily treated with Radiochemotherapie and there is a lack of better treatment option. However this study should be rolled out as a prospective double-blind clinical trial to confirm those findings. 

Following point should be addressed:

1.) The References show an error message in the text, so it is difficult to check the literature.

2.) Some abbreviations are not explained in the text (e.g. UFUR in the simple summary or PSM in the main text)

3.) Please provide detailed information and arguments why you have chosen a Propensity Score Matching Analysis. Which parameters were used to do this matching? Also would it be good to have a anonymized Table with each Patient-ID and their clinical details (e.g. as suppl. Table).

Author Response

Thanks for your kind review. Here are our responses. Meanwhile, a prospective double blinded phase III study is ongoing to confirm our results.

1.) The References show an error message in the text, so it is difficult to check the literature. 

Thanks for your review. We had carefully checked all the references.

2.) Some abbreviations are not explained in the text (e.g. UFUR in the simple summary or PSM in the main text) 

Thanks for your review. We had carefully checked all the abbreviations and corrected them.

3.) Please provide detailed information and arguments why you have chosen a Propensity Score Matching Analysis. Which parameters were used to do this matching? Also would it be good to have a anonymized Table with each Patient-ID and their clinical details (e.g. as suppl. Table).

Thank you for your suggestion. As mentioned in “Statistical Analysis” L3-5, in order to reduce the selection bias, patients with CCRT-UP were matched with those with CCRT-P by propensity score matching (PSM) according to age, gender, performance status, renal function and clinical stage. After PSM, a total of 88 patients were enrolled into our study. We think it is really difficult to have a table to present the clinical details of these 88 patients. Table 1 had already summarized all the clinical information of these patients.

Reviewer 3 Report

The aim of this manuscript is to investigate the oncologic outcomes of platinum plus tegafur-uracil versus platinum alone during CCRT in patients with non-metastatic NPC. A propensity score matching analysis was performed to diminish selection bias. This manuscript is clearly organized. This is a very important and interesting topic. However, it needs minor revisions to make it acceptable to cancers.

Main contents

L111-114.

In this study, the treatment regimen for CCRT-UP is " For CCRT-UP, patients received a 1-week cycle of cisplatin 35mg/m2 or carboplatin AUC 2 on day 1 of each cycle, plus oral UFUR 200mg twice per day and oral leucovorin 100mg twice per day for 7 days of each cycle during CCRT, followed by UFUR for 1 year or until disease progression or unacceptable toxicity.

There is no reference in the article that shows the efficacy of this regimen.

In the Discussion section, there are references to CF in a phase III study and raltitrexed (Saiweijian®) plus cisplatin (SP regimen) in a multicenter open label randomized study. .

Additional references showing the efficacy of CCRT-UP are needed.

If there is no evidence of reference for this regimen, a phase I study to determine the dose setting of the drug is needed.

Author Response

Thanks for your kind review. Here are our responses.

L111-114.  In this study, the treatment regimen for CCRT-UP is " For CCRT-UP, patients received a 1-week cycle of cisplatin 35mg/m2 or carboplatin AUC 2 on day 1 of each cycle, plus oral UFUR 200mg twice per day and oral leucovorin 100mg twice per day for 7 days of each cycle during CCRT, followed by UFUR for 1 year or until disease progression or unacceptable toxicity. There is no reference in the article that shows the efficacy of this regimen. In the Discussion section, there are references to CF in a phase III study and raltitrexed (Saiweijian®) plus cisplatin (SP regimen) in a multicenter open label randomized study. Additional references showing the efficacy of CCRT-UP are needed. If there is no evidence of reference for this regimen, a phase I study to determine the dose setting of the drug is needed.

Thank you for your review. Our study is the first study with great novelty regarding the treatment efficacy of CCRT-UP. Given that CCRT-UP is a novel regimen, there is no reference at present. Actually, a randomized double blind phase III study is ongoing to confirm our results. Tegafur-uracil (UFUR) is widely used in Asia and Europe for decades in treatment of patients with head and neck cancer [1], nasopharyngeal cancer [2], gastric cancer [3], colorectal cancer [4] and lung cancer [5]. The dose setting of UFUR is the recommended dosage. Thus, a phase I study to determine the maximal tolerated dose is not necessary.

  1. Huang WY, Ho CL, Chao TY, Lee JC, Chen JH. Oral tegafur-uracil as a metronomic therapy in stage IVa and IVb cancer of the oral cavity. Am J Otolaryngol. 2021 Nov-Dec;42(6):103156. doi: 10.1016/j.amjoto.2021.103156. Epub 2021 Jun 25. PMID: 34242883.
  2. Chen JH, Huang WY, Ho CL, Chao TY, Lee JC. Evaluation of oral tegafur-uracil as metronomic therapy following concurrent chemoradiotherapy in patients with non-distant metastatic TNM stage IV nasopharyngeal carcinoma. Head Neck. 2019 Nov;41(11):3775-3782. doi: 10.1002/hed.25904. Epub 2019 Aug 22. PMID: 31435974.
  3. Oba K. Efficacy of adjuvant chemotherapy using tegafur-based regimen for curatively resected gastric cancer: update of a meta-analysis. Int J Clin Oncol. 2009 Apr;14(2):85-9. doi: 10.1007/s10147-009-0877-4. Epub 2009 Apr 24. PMID: 19390937.
  4. Ishihara S, Hayama T, Yamada H, Nozawa K, Matsuda K, Watanabe T. Benefit of tegafur-uracil and leucovorin in chemoradiotherapy for rectal cancer. Hepatogastroenterology. 2011 May-Jun;58(107-108):756-62. PMID: 21830385.
  5. Watanabe K, Toi Y, Nakamura A, Chiba R, Akiyama M, Sakakibara-Konishi J, Tanaka H, Yoshimura N, Miyauchi E, Nakagawa T, Igusa R, Minemura H, Mori Y, Fujimoto K, Matsushita H, Takahashi F, Fukuhara T, Inoue A, Sugawara S, Maemondo M; North Japan Lung Cancer Study Group, Sendai, Japan. Randomized phase II trial of uracil/tegafur and cisplatin versus pemetrexed and cisplatin with concurrent thoracic radiotherapy for locally advanced unresectable stage III non-squamous non-small cell lung cancer: NJLCG1001. Transl Lung Cancer Res. 2021 Feb;10(2):712-722. doi: 10.21037/tlcr-20-721. PMID: 33718016; PMCID: PMC7947416.